# Effectiveness of Interventions and Behaviour Change Techniques for Improving Dietary Intake in Young Adults: A Systematic Review and Meta-Analysis of RCTs

**DOI:** 10.3390/nu11040825

**Published:** 2019-04-11

**Authors:** Lee M. Ashton, Thomas Sharkey, Megan C. Whatnall, Rebecca L. Williams, Aaron Bezzina, Elroy J. Aguiar, Clare E. Collins, Melinda J. Hutchesson

**Affiliations:** 1School of Health Sciences, Faculty of Health and Medicine, University of Newcastle, Callaghan 2308, Australia; lee.ashton@newcastle.edu.au (L.M.A.); thomas.sharkey@uon.edu.au (T.S.); megan.whatnall@uon.edu.au (M.C.W.); rebecca.williams@newcastle.edu.au (R.L.W.); aaron.bezzina@uon.edu.au (A.B.); clare.collins@newcastle.edu.au (C.E.C.); 2Priority Research Centre in Physical Activity and Nutrition, University of Newcastle, Callaghan 2308, Australia; 3Department of Kinesiology, School of Public Health and Health Sciences, University of Massachusetts Amherst, MA 01003, USA; eaguiar@umass.edu

**Keywords:** behaviour change techniques, young adults, nutrition, systematic review, meta-analysis

## Abstract

Poor eating habits are common during young adulthood and influence chronic disease morbidity. This systematic review evaluates the effectiveness of interventions aiming to improve dietary intake among young adults and, identifies which behaviour change techniques (BCTs) are most effective. Six electronic databases were searched for RCTs published until October 2018, and evaluating behavioural interventions assessing change in dietary intake in young adults (17–35 years). Of the 18,779 articles identified, 54 were included. Forty studies focused on fruit and/or vegetable intake, of which 63% showed a significant between-group difference in favour of the intervention group. Meta-analysis (*n* = 17) demonstrated a significant increase in fruit and vegetable intake of +68.6 g/day after three months of intervention and +65.8 g/day for interventions >3 months when compared to control. A meta-analysis (*n* = 5) on total energy intake found no significant differences between groups. The BCTs with the highest effectiveness ratio were habit formation (100%), salience of consequences (83%) and adding objects to the environment (70%). The review highlights the potential of behavioural interventions to improve young adults’ fruit and vegetable intake but was less convincing for other dietary outcomes. Due to the lack of studies including each BCT, the BCTs imperative to success could not be identified.

## 1. Introduction

Globally, young adults have some of the unhealthiest eating habits compared to other age groups. In a systematic assessment of diet quality among adults in 187 countries, those aged 20–29 years had the lowest diet quality score (mean 36 ± 10 out of a maximum of 100 with higher score indicating better diet quality) compared to any other age group [1]. Low diet quality indicates lower intakes of healthier items from the core food groups such as fruits and vegetables, wholegrains and key nutrients including omega-3s and dietary fibre, and higher intakes of energy-dense, nutrient-poor foods such as processed meats, sugar-sweetened beverages (SSB), saturated and trans fats and sodium.

One of the influencing factors to these poor dietary choices in this age group are the major life transitions which take place and have resulted in instability in habitation, relationships and employment status [2]. During young adulthood, the key transitions include changes in the living situation, with individuals often moving out of the family home to live independently; changes in the social environment and social influences; moving from family dependence towards stronger peer networks and partner relationships; changes in work and/or study from secondary to tertiary education and into employment or unemployment; and changes in financial status with young adults becoming financially independent [3]. These key life transitions are associated with negative changes to the diet [4] and present an opportunity to target improvement in eating habits during this life stage. Poor diet quality is one of the leading causes of mortality and disability worldwide [5], and therefore intervening while adults are still young is central to the prevention of noncommunicable chronic diseases and promotion of healthy ageing [6]. 

The poor dietary intake during young adulthood coincides with rapid weight gain; with average weight increases in the range of 0.5kg to 1kg per year from early to mid-adulthood [7,8]. This could lead to detrimental effects on health; for example, the Health Professionals Follow-up Study [9] (*n* = 118,140) identified that for every additional 5kg of weight gained from 21 years of age there were 65 additional cases (per 100,000 person-years) of mortality from age 55 years onwards in males and 77 additional cases in females. Also, there were 496 additional cases of hypertension in males and 459 in females, 151 cases of type 2 diabetes in males and 143 in female, 46 cardiovascular disease cases in males and 37 in females and eight additional cases of cancer in males and 37 in females at age 55 years onwards.

There has been strong support recently for successful interventions to improve diet and prevent weight gain among young adults. Specifically, a 2017 Journal of American Medical Association editorial concluded: “Reducing and preventing obesity and excessive weight gain in young adults provide a new target and one that could offer an effective transgenerational approach for prevention” [10]. Accordingly, there has been a surge in research based healthy lifestyle programs which target dietary intake among young adults in recent years. There have been several recent systematic reviews of lifestyle interventions with dietary outcomes among young adults. However, these are limited by a refined focus: on the population—specific to University students [11,12]; on delivery mode—specific to *e*- or *m*health [13,14,15]; or on the outcome—specific to weight and anthropometry outcomes [16,17]. This presents an opportunity to explore the effectiveness in all potential interventions involving young adults and also determine which aspects of these interventions are contributing to effectiveness. A Behaviour Change Technique (BCT) is defined as an “active ingredient” of an intervention [18]. Precise specification of these active ingredients and intervention features of diet in interventions will help build cumulative evidence towards delivering effective replicable interventions [19]. However, to date no systematic review has been conducted to identify BCTs associated with healthy eating in young adults. Therefore, the aims of this review were to

evaluate the effectiveness of interventions aiming to improve dietary intake among healthy young adults (aged 17–35 years) andidentify the BCTs used in these interventions and determine which are most effective.

## 2. Materials and Methods 

### 2.1. Protocol and Registration

This systematic review and meta-analysis was conducted as per the PRISMA guidelines [20], and used a predefined protocol registered with PROSPERO (CRD42017075795) [21]. 

The full systematic review described in the protocol evaluates interventions targeting nutrition, physical activity (PA) and/or the treatment or prevention of obesity among young adults. This paper presents results for those interventions aiming to improve dietary intake only.

### 2.2. Eligibility Criteria

#### 2.2.1. Types of Participants

Young adults aged 17–35 years were included to align with the international definition of young adulthood [22]. The broad age range ensured a wider scope to obtain a greater number of studies. The intention of this review is to provide recommendations for the general healthy young adult population. Therefore, studies with participants from groups with diagnosed conditions linked to obesity risk factors (e.g., type 2 diabetes) or from special populations (e.g., severe mental illness, eating disorders, elite athletes and pregnant women) were excluded.

#### 2.2.2. Types of Interventions

Behavioural interventions (focusing on diet, PA and/or treating or preventing obesity), which assessed change in dietary intake, were included. Dietary change is rarely managed in isolation which is why interventions targeting PA and obesity were also considered. Interventions were required to be designed specifically to promote behaviour change. As such, studies which primarily investigated the acute impact of weight loss on other clinical biomarkers (e.g., insulin) were excluded. In addition, supervised, controlled exercise programs primarily examining the impact of exercise on clinical biomarkers and fitness and not designed to change lifestyle behaviours were excluded, as were obesity interventions involving bariatric surgery or anti-obesity medications.

#### 2.2.3. Types of Comparators

Any comparators were considered for inclusion. This includes comparison with no intervention (e.g., waitlist control) and/or compared to active treatments.

#### 2.2.4. Type of Outcome Measures

Any measures to assess effectiveness of interventions on dietary intake as the primary outcome in a dietary intervention or as a secondary outcome in PA and/or obesity interventions (e.g., reporting energy, nutrient or food group intake, dietary patterns or diet quality). A measurement at baseline and a minimum of one postintervention time point were required for inclusion.

#### 2.2.5. Types of studies 

Only randomised controlled trials (RCTs) including pilot and feasibility RCTs were considered. 

### 2.3. Information Sources and Search 

Six electronic databases were searched: MEDLINE (Ovid), EMBASE (Ovid), PsycINFO (Ovid), Science Citation Index (WoS), Cinahl (EbscoHost) and Cochrane Library (Wiley) from the date of inception to October 2018 (Appendix A). The search was limited to studies published in English Language and consisted of focused ‘text word’ searches and subject heading searches (MeSH). Linked papers to the relevant RCTs (i.e., protocol papers, recruitment papers, process evaluation papers and those presenting outcomes at different time points) were also collected. The reference list of retrieved papers and pertinent systematic reviews were also searched. A citation search of included papers using Scopus was conducted to identify any additional papers meeting the criteria. The Scopus citation search involved screening the articles in the ‘cited by’ list for each of the included articles.

### 2.4. Study Selection

The title, abstract and keywords of all identified papers were assessed by two independent reviewers (L.M.A. and M.J.H., M.C.W., C.E.C., T.S., R.L.W. or E.J.A.). The full text of records of those which appeared to be relevant or unclear were then retrieved and assessed by two independent reviewers to determine inclusion or exclusion (L.M.A. and M.C.W., T.S. or R.L.W.), with a third reviewer resolving any disagreements (M.J.H.). Where papers did not provide sufficient detail to determine eligibility (e.g., age range not provided) the corresponding author was contacted to confirm if the inclusion criteria were met (*n* = 32). If the author did not respond, the study was subsequently excluded. Reasons for exclusion were recorded for all full text retrieved papers deemed ineligible for inclusion in this review. 

### 2.5. Risk of Bias 

The Cochrane Collaboration’s Tool for assessing risk of bias [23] was completed by two independent reviewers (L.M.A. and M.J.H., A.B., R.L.W. or M.C.W.) and a third reviewer consulted in case of disagreement (M.J.H. or T.S.). 

### 2.6. Data Extraction 

Data were extracted by one reviewer (T.S) and checked by a second reviewer (L.M.A. or M.J.H., A.B., R.L.W or M.C.W). This included study characteristics (e.g., authors, title and date of publication, country and duration), study participants (e.g., number of participants, age and gender), experimental conditions (e.g., number of study arms, description of intervention and comparator) and study outcomes (e.g., dietary outcomes reported and between group differences in results). 

### 2.7. Coding of BCTs

The 93-item Behaviour Change Taxonomy v1 [18] was used to assess the inclusion of BCTs in intervention content. The Behaviour Change Taxonomy provides a standardised method for reporting techniques used in behaviour change interventions, allowing for between study comparisons and effective intervention features to be identified. BCTs were coded by two independent reviewers (LMA and MCW with 100% agreement). All reviewers completed an online BCT course prior to coding http://www.bct-taxonomy.com/ and a BCT was only coded when there was clear evidence of inclusion. When interventions targeted more than one behaviour, the techniques and results were recorded specifically for the diet behaviour. The BCTs in the intervention and control groups were identified separately. Only BCTs present in the intervention and absent in the control condition were recorded. This approach was used to explain the difference in effect as emphasised by Peters and colleagues [24], and used by Samdal and colleagues [25]. In studies with multiple intervention arms, the BCTs and results were extracted for each active arm compared to control. For those studies with only active intervention arms (i.e., new treatment versus old treatment), the group deemed as primary (established in aims or methods) was considered as the intervention group and the other was considered as the control/comparison.

### 2.8. Synthesis of Results and Analytic Strategy 

#### 2.8.1. Narrative Summary

Due to the heterogeneity in dietary outcomes, the majority of results are described in narrative form. There were sufficient comparable studies which assessed change in total energy intake and/or fruit and vegetables in intervention versus control group. Therefore, these were included in the meta-analysis. 

#### 2.8.2. Meta-Analysis 

Meta-analyses of change in total energy intake (TEI) (kJ/day) and change in fruit and vegetable intake (g/day) across studies were conducted to assess change in reported intakes at each time point for both intervention and comparator/control groups. The meta-analysis was conducted using R statistical software (V 3.5.1, Vienna, Austria) using the Metafor package (V 2.0, Vienna, Austria). To standardise total energy intake, studies reporting calories per day were converted to kilojoules per day by multiplying by 4.184. When fruit and vegetables were expressed in either portions per day or serves per day, we used a standardisation procedure from previous meta-analyses [26,27,28], whereby the exposure level was transformed into a grams per day by multiplying this quantity by 106g [26,28]. The five studies which reported fruit and vegetables in cups per day were from the U.S., and therefore we used 160g as a cup equivalent size because the U.S. serving of one-half cup of vegetables or a medium-size piece of fruit was taken to be equal to a portion of 80g [27]. 

For studies reporting intake of fruit and vegetables separately, we converted the results as follows; the means for fruit and for vegetable were summed to give a total fruit and vegetable mean. For the standard deviation (SD) and sample size, a conservative approach was taken assuming the Standard Error’s (SE) for each mean were different. SD combined score was determined as [n1.SE1^2^ + n2.SE2^2^]^½^ and the sample size was as reported for the study. If SDs rather than SEs were reported, the SDs were used in place of the SEs and n1, n2 was set at 1. 

For each study, the effect at baseline and each time point, expressed as months post baseline, was estimated as the mean difference (mean intervention group − mean control group) using the unbiased option for variance estimates. Uncertainty in the mean estimates was expressed as 95% confidence intervals. To account for multiple measures per study a series of multilevel models were investigated with nested random effects top level being study, then treatment type and time period using REML estimation. The final model contained study and time as nested random effects as the treatment type random effect had zero variance. Moderator variables examined as fixed effects in the model were treatment type (simplified into three groups for fruit and vegetables and two groups for TEI), and time was treated as a categorical variable in two versions (as originally reported with 15 different levels for fruit and vegetables and eight levels for TEI) and a simplified grouped form with baseline, up to three months and greater than three months. The interaction between these two was tested. As the time random effect was similar in size to the study random effect there was substantial correlation between time periods, so the moderator effect for the three-group version of time was estimated at each of the two follow-up time periods as difference from baseline. Residual plots were used to examine homogeneity of variance and normality assumptions. To assess for publication bias, funnel plots were produced and visually inspected and supported with rank correlation tests for funnel plot asymmetry.

### 2.9. Effectiveness of BCTs

In parallel with similar reviews [29,30], we calculated a percentage effectiveness ratio to identify the BCTs associated with effective interventions, whereby the number of times the technique was a component of an effective intervention divided by the total number of times the technique was a component of an intervention. An effective intervention was defined as one where the change in one or more nutrition outcomes was positive and statistically significantly different from baseline, compared with control, or if no control comparator, compared with another active intervention. Only the BCTs which were identified in a minimum of five studies were included in the analysis to avoid inflation of results from those BCT’s used on few occasions. A plot of effectiveness with 95% credible intervals from a Bayesian approach (beta posterior from a binomial likelihood and conjugate uniform beta prior) was implemented. As there were 24 BCTs that were identified in at least five studies, an overall test of significance was carried out using a contingency table between type and number of BCTs and effectiveness using Monte–Carlo exact chi-squared test (SPSS version 25, Armonk, New York, USA). 

## 3. Results

### 3.1. Description of Included Studies

Of the 18,779 manuscripts identified, a total of 54 individual studies (and a total of 78 papers) were included in the review (Figure 1). Study characteristics are summarised in Table 1, with detailed study characteristics provided in Appendix A. Of the 54 included studies, 17 were eligible for meta-analysis of fruit and vegetable intake [31,32,33,34,35,36,37,38,39,40,41,42,43,44,45,46,47] and five were eligible for meta-analysis of TEI [48,49,50,51,52].

Interventions to improve eating habits among young adults has escalated in recent years, with over half of the included studies published from 2014 to 2018 (*n* = 29, 54%). In addition, most studies were from the US (*n* = 30, 55.5%), in a college/university setting (*n* = 37, 68.5%) and predominantly in white/Caucasian populations (*n* = 32, 59%). The included studies had a total of 16,383 participants (median: 162, range: 37 to 2343) and the mean age of participants was 20.2 years with 57% (*n* = 31) within the age category of 17–≤ 25 years. Most studies included both male and female samples (*n* = 39, 72%), and of these the average proportion of males was 32%. Over one-third of the interventions were multicomponent (*n* = 21, 39%), and there was an equal proportion of studies which solely used either *e*health or face-to-face as the intervention delivery (both *n* = 16, 30%). Over half of the included studies had dietary change as a primary outcome (*n* = 32, 59.3%), while dietary change was a secondary outcome for the remaining 22 (40.7%) studies. Studies where dietary change was a secondary outcome primarily focused on obesity/ weight gain prevention (*n* = 11, 20.4%), obesity treatment (*n* = 8, 14.8%) or PA (*n* = 3, 5.6%). In the studies with dietary change as a primary outcome, most focused on fruit and/or vegetable intake (*n* = 26 out of 32, 81%), while the remaining focussed on diet quality score (*n* = 3, 9.4%), micronutrients (*n* = 2, 6.3%) or percent of energy coming from fat (*n* = 1, 3.1%). 

Twenty-five (46%) studies used a food frequency questionnaire (FFQ) for assessment of dietary intake [31,32,37,38,41,42,44,45,48,50,53,54,55,56,57,58,59,60,61,62,63,64,65,66], 17 (32%) used a specific nutrient/food/diet behaviour questionnaire [33,34,35,36,39,40,43,47,67,68,69,70,71,72,73,74,75], five (9%) used a food record [46,49,52,76,77] and five (9%) used a dietary recall [51,52,78,79,80]. For the recalls, three used a 24-hour recall [51,79,80], one used a 3-day recall [52] and one used a 7-day recall [78]. Two studies used more than one method of dietary assessment [81,82]

The total number of study arms was 133 (range: 2 to 6 arms), with the majority being two-arm interventions (*n* = 38, 70%). The mean intervention duration across the studies was 4.2 months (range: single session to 24-months), with the majority being short term between one single session and ≤3-months (*n* = 40, 74%). The mean length of follow-up from end of intervention was 1.8 months (range: no follow-up to 23-months), with the majority having between no follow-up and ≤3-months (*n* = 45, 83%). Finally, there was a mean retention rate of 78% (range 22% to 98%) at the end of the intervention and a mean retention rate of 66% (range 11% to 97%) at the longest follow-up point. 

### 3.2. Risk of Bias

The risk of bias assessment is summarised in Figure 2. There was a low risk of bias for incomplete outcome data (attrition bias), as over half (*n* = 28, 52%) adequately described study attrition and any exclusions from the analysis. Almost half (*n* = 26, 48%) of the included studies described suitable generation for the allocation sequence, yet most (*n* = 38, 70%) failed to describe the method for allocation concealment. There was a high or unclear risk of bias for most studies for blinding of participants (*n* = 49, 91%), blinding of those delivering the intervention (*n* = 37, 68%) and blinding of outcome assessors (*n* = 34, 63%). Most studies provided insufficient detail (*n* = 39, 72% unclear) to determine if they were free of selective outcome reporting.

### 3.3. Dietary Outcome: Total Energy Intake 

#### 3.3.1. Total Energy Intake: all Included Studies 

When all studies were considered, there were a total of twelve studies which measured change in TEI (none as primary outcome), reported as either Calories or kilojoules per day [48,49,50,51,52,53,57,58,64,77,79,83]. This includes the five studies from the meta-analysis and a further seven studies that reported TEI as an outcome, but results were not comparable and therefore not included in the meta-analysis. Of these, only one study (with a focus on weight gain prevention) [50] reported a significant between group difference in TEI at 4-months (*p* = 0.014) with a reduction of 326 kcal/day in the intervention group (comprised of group-based face-to-face lecture and lab sessions) compared to +72 kcal/day in the usual-lifestyle control group. This effect was not significant at 16-months. 

#### 3.3.2. Total Energy Intake: Meta-Analysis

Meta-analysis of total energy intake (TEI) in five studies [48,49,50,51,52] for interventions targeting healthy eating among young adults examined two moderator effects. There was no significant time effect (LRT χ2(4) = 0.47, *p* < 0.790) showing a mean decrease in TEI relative to baseline −132.9 kJ/day up to 3 months (95% CI: −824.2, 558.4) and −300.2 kJ/day for >3 months (95% CI: −1002.7, 402.4) (Figure 3). There was no significant treatment type effect, Wald χ2(1) = 0.48, *p* = 0.4904). The funnel plot (Appendix A) demonstrated no symmetry, indicating there was no evidence of publication bias to higher values, a nonparametric correlation test supported this, Kendall’s tau ‒0.19, *p* = 0.39. Plots of the means for the effects are in Appendix A, while model diagnostics are satisfactory and are in Appendix A. The forest plots showing mean difference) 95% confidence interval) over time (months) are in Appendix A. 

### 3.4. Dietary Outcome: Fruit and Vegetables

#### 3.4.1. Fruit and Vegetable Intake: All Included Studies

A total of 40 studies measured fruit and/or vegetable intakes [31,32,33,34,35,36,37,38,39,40,41,42,43,44,45,46,47,53,54,56,57,59,60,62,66,67,68,69,70,71,72,73,74,75,76,77,78,79,81]. Of these, statistically significant between-group increases in fruit and/or vegetable intakes were reported in 25 studies, as shown in Appendix A [31,34,35,36,38,40,41,42,46,50,53,54,57,59,60,62,67,70,72,73,75,77,78,81]. The majority of those effective (18 out of 25) had a primary focus to improve fruit and/or vegetable intake [31,34,35,36,40,41,42,54,59,62,70,72,73,75,78,81]. 

Fruit and vegetable intake were reported as (i) serves per day in 12 studies [31,35,39,40,44,47,54,68,69,71,74,81], with five demonstrating a positive between-group difference in favour of the intervention group [31,35,40,44,54]; (ii) as cups per day in five studies [33,36,38,41,45], with three demonstrating a positive significant between group difference [36,38,41]; (iii) as portions per day in three studies [34,66,73], with two of these showing a positive significant between-group difference [34,73]; (iv) as percent meeting recommendations in two studies [37,72], with one demonstrating a significant improvement in intervention group compared to usual lifestyle control group [72]; (v) as frequency in one study [62], showing a significant increase in all four active intervention arms at four weeks when compared to control; and (vi) the unit was not provided in one study [56] and demonstrated no difference between groups at either six weeks or three months. 

There were 18 studies that reported fruit intake separately from vegetable intake [32,37,42,43,46,53,54,57,59,60,67,70,75,76,77,78,79], of these, eight studies demonstrated a significant increase in fruit intake in the intervention group compared to control [46,54,59,60,67,70,75], while 10 studies demonstrated a significant increase in vegetable intake compared to control [42,46,53,57,59,60,67,77,78]. 

#### 3.4.2. Fruit and Vegetable Intake: Meta-Analysis 

Meta-analyses of fruit and vegetables in 17 studies [31,32,33,34,35,36,37,38,39,40,41,42,43,44,45,46,47,81] for interventions targeting healthy eating among young adults examined two moderator effects. There was a significant time effect (LRT χ^2^(5) = 16.96, *p* < 0.001) showing a mean increase in fruit and vegetable intake relative to baseline 68.6 g/day up to 3 months (95% CI: 36.6, 100.5) and 65.8 g/day for >3 months (95% CI: 27.6, 104.0) (Figure 4). There was no significant treatment type effect, Wald χ^2^(2) = 0.42, *p* = 0.8091) nor an interaction between treatment type and time Wald χ^2^(4) = 0.318, *p* = 0.53). The funnel plot (Appendix A) demonstrated symmetry, indicating there was publication bias to higher values; a nonparametric correlation test supported this, Kendall’s tau 0.35, *p* < 0.001. Removing the study by Chang (2010) with the two largest values in the funnel plot did not change the bias tau 0.30, *p* = 0.003. Plots of the means for the effects are in Appendix A, while the model diagnostics are satisfactory and in Appendix A. The forest plots showing mean difference (95% confidence interval) over time (months) are in are in Appendix A. Of the 17 studies in the meta-analysis, 11 had a primary focus to change fruit and vegetable intake [31,34,35,36,39,40,41,42,44,45,46].

### 3.5. Dietary Outcome: Nutrient-Dense Foods (Other Than Vegetables and Fruit)

A total of 15 studies assessed the change in intakes of at least one nutrient dense food [38,42,46,48,49,53,55,56,57,63,65,77,79,80,81], and this was reported as diet quality score rating [53,63,65,77,80], as individual food groups (i.e., wholegrains, dairy, fish, meat, cereals and staple foods) [38,42,46,77,79], as a percentage of total daily energy intake [57] or as a food high in a particular micronutrient (i.e., calcium and vitamin D) [49,55] or dietary fibre [48,56,77,79,81]. A statistically significant between-group change that favoured the intervention group in at least one nutrient-dense food was evident in five of the 15 studies assessing nutrient dense foods [46,49,57,65,79]. 

#### 3.5.1. Diet Quality Score Rating

Statistically significant improvements in diet quality score was evident in one study (out of five) [65]. This study has a primary focus to improve overall diet quality and included a group-based face-to-face intervention based on the Health Belief Model. The intervention group significantly (*p* < 0.001) increased Diet Behaviour Score (+0.61, *p* < 0.001) when compared to usual lifestyle control group (−0.46, *p* < 0.02) at 1-month. 

#### 3.5.2. Individual Food Groups

A significant between-group difference in individual food groups was evident in one study (out of five) [46]. This nutrition intervention provided nutrition information and focused on increasing availability of vegetables and whole grain intake within conscription based young male participants. There was a significant (*p* < 0.001) increase in whole grain bread intake in the intervention group (+46g/day) compared to control (−12g/day) at 5-months. 

#### 3.5.3. Percentage of total Daily Energy Intake 

The one study which explored the effectiveness of an *e*health weight loss program (Be Positive Be Health*e*) for young women on percentage energy from core foods found a significant improvement in the intervention group when compared to control (mean difference: +9.4%, *p* = 0.02) at six months. 

#### 3.5.4. Micronutrients & Dietary Fibre 

Statistically significant improvements in specific micronutrient and dietary fibre rich foods were evident in two studies (out of seven) [49,79]. Specifically, participants in the intervention group consisting of gain-framed materials relating to nutritional information of calcium-rich foods increased their intake of calcium rich foods at six months and 12 months when compared to the standard care control group [49]. A statistically significant increase in fibre intake was evident in one study (out of five) [79]. The intervention group, which consisted of a smartphone application for weight loss combined with SMS from a health coach, had a statistically significant increase in fibre intake at 3-months when compared with control group who received a one-time counselling session (*p* = 0.049). 

### 3.6. Dietary Outcome: Energy-Dense Beverages 

Eight studies assessed the change in energy-dense beverage intake [32,38,43,46,61,67,69,76] with seven of these targeting sugar sweetened beverages [32,38,43,61,67,69,76] and one targeting fruit juice intake [46]. Statistically significant between-group reductions in energy-dense beverage intake were evident in three studies [43,46,67].

#### 3.6.1. Sugar Sweetened Beverages (SSBs)

Two studies (out of seven) demonstrated a significant reduction in SSB intake when compared to control [43,67]. Specifically, the TXT2BFIT weight gain prevention program was delivered via multiple *e*Health components (i.e., SMS, email, mobile app and website) in young Australian adults for 12 weeks. The odds that the TXT2BFIT intervention group improved intake compared to the control group were significantly greater for SSBs at the end of the program (*p* = 0.024) and at 9-month follow-up (*p* = 0.018) [67]. Similarly, an 8-week intervention in exclusively young women delivered via print and group-based face-to-face sessions found a significant between-group difference in SSB (drinks/day) at the end of the program (*p* = 0.03). The SSB intake decreased in the intervention group (−0.53 drinks/day) and increased in the control group (+0.50 drinks/day) [43].

#### 3.6.2. Fruit Juice

One study reported change in fruit juice intake and a small but significant between-group effect was evident (*p* = 0.001) [46]. Although both groups increased fruit juice intake at 5-months, the intervention group which consisted of nutrition education and changes to the food environment within conscription based young male participants had a smaller increase (+25 g/day) when compared to control (+42 g/day). 

### 3.7. Dietary Outcome: Energy-Dense Nutrient-Poor (Ed-Np) Foods

A total of 12 studies assessed the change in intakes of at least one ED-NP food [31,32,37,43,45,53,57,61,67,69,76,79], and this was reported as percentage of total daily energy intake [45,53,57], as takeaway/fast food [32,57,61,67,69], as individual foods/food groups (i.e., hot chips, sweets, junk food and fried food) [31,37,43,76] and salt and sugar [79]. A statistically significant between-group change that favoured the intervention group in at least one ED-NP food was evident in four of the 12 studies [53,57,61,67].

#### 3.7.1. Percentage of Total Daily Energy Intake

Two studies (out of three) demonstrated a significant reduction in percentage energy from ED-NP foods when compared to control [53,57]. Both of these studies were gender-targeted. The HEYMAN program for young men was delivered via *e*health (website, mobile app, social media and wearable device) and face-to-face sessions. Those in the HEYMAN intervention had a reduction of 9.8% (95% CI: −13.3, −6.2) in EDNP foods/day, compared to a reduction of 2.6% (95% CI: −6.2, 1.1) in the control group. The ‘Be Positive Be Health*e*’ weight loss program for young women was delivered via *e*health components (website, social media, smartphone application, email and text messages). Those in the intervention group had a reduction of 8.3% (95% CI: −13.7, −3.0) in EDNP/day, compared to an increase of 0.9% (95% CI: −1.2, 1.2) in the control group. 

#### 3.7.2. Takeaway/Fast Food

Two studies (out of five) demonstrated a significant reduction in takeaway/fast food intake when compared to control [61,67]. The odds that the TXT2BFIT intervention group improved intake compared to the control group were significantly greater for frequency of weekly take-out meals at the end of the program (*p* = 0.013) and at 9-month follow-up (*p* = 0.010) [67]. The CHOICES study (Choosing Healthy Options in College Environments and Settings) tested effects of a technology-integrated, young adult weight gain prevention intervention [61]. Those assigned to the CHOICES intervention group had a significant decrease in fast food consumption compared to control at 24-months (*p* = 0.007).

#### 3.7.3. Individual Foods/Food Groups 

Four studies considered the effect of a healthy eating intervention on hot chips (serves/week) [31], sweets/lollies (serves/week) [31], ED-NP foods (serves/day), fried food (serves/day) [76] or junk food (serves/day) [43]. None of these studies demonstrated a significant positive intervention effect on any of these outcomes. 

#### 3.7.4. Salt or Sugar 

One obesity treatment study explored the effect of an intervention consisting of a smartphone application combined with SMS from a health coach on sodium intake, added sugar intake and total sugar intake [79] in overweight and obese young adults. There were no significant intervention effects for any of these outcomes. 

### 3.8. Dietary Outcome: Macronutrients 

A total of 15 studies assessed change in macronutrient content (carbohydrates, proteins and/or fat) [33,38,45,47,48,50,52,56,58,66,77,79,81,82,83]. This was reported as percentage of daily energy intake in 11 studies [33,38,45,48,50,58,66,77,81,82,83], as daily grams in three studies [50,52,77], as frequency in one study [47] and unit was not provided in two studies [56,79]. Of these, a statistically significant between-group change in macronutrient intakes that favoured the intervention group was reported in four studies [38,45,50,82]. 

Specifically 15 studies explored change in dietary fats [33,38,45,47,48,50,52,56,58,66,77,79,81,82,83], eight for carbohydrate intake [47,48,50,52,58,77,79,83] and seven for protein intake [48,50,52,58,77,79,83].

#### 3.8.1. Changes in Intake of Dietary Fats

Four studies (out of 15) reported a significant intervention effect in dietary fat intake [38,45,50,82]. Specifically, one weight gain prevention study reported a between-group difference at the end of the program at 10 weeks (−0.9% vs. +0.01, *p* < 0.01) and at 15-month follow-up (−0.8% vs +0.01%, *p* < 0.01) in percent energy from fat [38]. Another study which was also weight gain prevention intervention reported a between-group difference at 16-month follow-up (−0.5% vs +2.4%, *p* < 0.01) in percent energy from fat [50]. The same study found no intervention effects at the end of the program (four months), nor were there any effects for total fat (g/day) at any time point [50]. An *e*Health intervention consisting of email and website to improve diet and physical activity had a significant linear time by treatment interaction for percent energy from saturated fat (F_1,372_ = 3.94, *p* = 0.048) at 12 weeks and 24 weeks [45]. Finally, in a study exploring the effect of three intervention arms, there was a significant between group difference for change in percentage kcal from fat at end of intervention (6 weeks) for intervention arm one which consisted of nutrition counselling and serum cholesterol feedback when compared to control (−3.2% vs. +0.2%, *p* = 0.02) [82]. No other significant between group differences were found between any of the groups. 

#### 3.8.2. Changes in Carbohydrate intake

One study (out of eight) reported a between-group decrease in total grams of carbohydrate at the end of the program at four months (−38.3g/day vs. +26.1g/day, *p* < 0.05), but this effect was not sustained at 16-month follow-up [50]. This study was a weight gain prevention intervention among university students delivered via group-based lectures and laboratory exercises with a focus on dietary intake, PA and weight. 

#### 3.8.3. Changes in Protein Intake

The same study which identified a change total grams of carbohydrate also reported a between-group decrease in total grams of protein at the end of the program at 4 months (−12.4 g/day vs. +26.1 g/day, *p* < 0.05) but this effect was not maintained at 16-month follow-up [50]. All the other studies (*n* = 6) which assessed change in protein intake did not establish any significant differences between groups.

### 3.9. Dietary Outcome: Diet Behaviour 

One study explored the effect of a healthy lifestyle program on diet behaviour [61]. The CHOICES weight gain prevention intervention (Choosing Healthy Options in College Environments and Settings) which was delivered via an academic course and a social network and support website looked at changes in breakfast eating (days/week) and weekly meals prepared at home. There were no significant differences between groups on either diet behaviour at four months, 12 months or 24 months. 

### 3.10. Behaviour Change Techniques 

#### Description of Behaviour Change Techniques Applied

Table 2 summarises the behaviour change techniques applied in the 70 active intervention arms, coded according to the Behaviour Change Taxonomy v1 [18]. Of the 93 BCTs, 55 were coded one or more times, with a total of 430 BCTs coded across the active intervention arms. Across all interventions arms, a median of five BCTs were employed (ranging from one to 25). The most frequently coded techniques included ‘Instruction on how to perform a behaviour’ (*n* = 45), ‘Goal setting behaviour‘(*n* = 32), ‘Information about health consequences’ (*n* = 29), ‘Action planning’ (*n* = 27) and ‘Social support (unspecified)’ (*n* = 24). 

### 3.11. Effectiveness of Behaviour Change Techniques

The percentage effectiveness ratio for the coded BCTs is displayed in Figure 5. Effectiveness was determined for 51 studies. For the remaining three studies, two had no difference in BCTs coded between intervention and control groups [71,77], while one publication [50] conducted two studies with the same BCTs utilised and the same outcome, and thus this was only recorded once. To provide a meaningful representation of effectiveness, only the BCTs identified in a minimum of five studies (*n* = 24) were included in the analysis [25].

Thirteen BCTs had an effectiveness ratio >50%. The BCTs with the highest effectiveness ratio were ‘Habit formation’ (effective in five out of five or 100% of interventions), ‘Salience of consequences’ (effective in five out of six or 83% of interventions), ‘Adding objects to the environment’ (effective in seven out of 10 or 70% of interventions), ‘Action planning’ (effective in 16 out of 27 or 59% of interventions) and ‘Prompts/cues’ (effective in seven out of 12 or 58% of interventions). 

An overall test of significance was carried out using a contingency table between type of BCT and effectiveness, there was no significant relationship, Monte–Carlo exact chi-squared test, χ2(23) = 16.7, *p* = 0.85, indicating that none of the BCTs were different to the overall mean value. Although ‘Habit formation’ had the highest percentage effectiveness ratio, many more studies are required before this could be confirmed as having greater effectiveness than others.

The percentage ratio of effective versus noneffective interventions by the number of BCTs was also explored (Figure 6). The results show no particular pattern as to whether fewer or more BCTs were associated with effectiveness. An overall test of significance using a contingency table showed there was no significant relationship, Monte–Carlo exact chi-squared test, χ2(15) = 7.9, *p* = 0.93. 

## 4. Discussion

This is the largest systematic review to date (*n* = 54 RCTs) that has assessed the effects of interventions aiming to improving dietary intake outcomes in young adults. It is also the first to explore the effectiveness of BCTs for improving dietary intake among this group. Predominantly, studies targeted fruit and/or vegetable intake as the primary outcome and a meta-analysis (*n* = 17 studies) demonstrated intervention participants to significantly increase intake of fruit and vegetables by +68.6 g/day up to three months and +65.8 g/day for >3 months when compared to control. A meta-analysis exploring change in TEI (*n* = 5 studies) showed no significant differences between groups. The narrative synthesis showed that interventions targeting fruit and/or vegetable intake were most effective (significant between group differences evident in 25 out of 40 studies or 63%), followed by energy-dense beverages (three out of eight or 38%), nutrient dense foods (five out of 15 or 33%) and macronutrients (four out of 15 or 27%). The BCTs with the highest percentage effectiveness ratio were habit formation, salience of consequences and adding objects to the environment. Despite this, the overall test of significance demonstrated no significant relationship between type of BCT and effectiveness and many more studies are required before these BCTs could be confirmed as having greater impact on effectiveness than others.

### 4.1. Effectiveness of Interventions on Dietary Outcomes

The modest increases in fruit and vegetable intake (mean difference of +68.6 g/day up to three months and +65.8 g/day for >3 months) is clinically important. In another meta-analysis among prospective cohort studies with 833,234 participants, the risk of all-cause mortality was decreased by 5% and cardiovascular disease by 4% for each additional serve a day of fruit and vegetables [84]. This is especially important as globally an estimated 710,000 coronary heart disease deaths, 1.47 million stroke deaths, 560,000 cancer deaths and 5.4 million premature deaths were attributable to a fruit and vegetable intake below 500 g/day in 2013 [85]. Increasing fruit and vegetable intake can also improve overall diet quality [86]. A systematic review and meta-analysis found increases in fruit and vegetable consumption increased micronutrient, carbohydrate and fibre intakes, but had no overall effect on energy intake [86]. 

When compared to other reviews in adults and children, the mean difference in fruit and vegetable intakes between intervention arms (approximately half serve/day) is lower. A systematic review of 34 behaviour-based interventions found an average increase in fruit and vegetable intake of +1.13 and +0.39 serves per day in adults and children [87], while another review in eight studies found a mean difference between arms of approximately one serve/day (mean: +133g/day, range 50g to 456g) [88]. This suggests that young adults are among the hardest group to implement positive behaviour change for dietary outcomes and corresponds with the global evidence which has identified this group as having the lowest diet quality compared to any other age group [1]. In parallel to this, results from the narrative synthesis demonstrated that a higher proportion of studies failed to identify a significant between-group change in favour of the intervention group for energy-dense beverages, nutrient dense foods (other than fruit and vegetables) and macronutrients. 

Additionally, the meta-analysis demonstrated no significant effect on TEI. Of the five studies in the meta-analysis, none had TEI change as a primary outcome, and only one study [52] aimed to reduce TEI for weight loss in overweight and obese participants. Also, three of the interventions were trying to maintain TEI for weight gain prevention [48,50,51], thus it is unlikely that the studies were powered to detect a change in TEI and may explain the minimal changes. When compared to results of the meta-analysis in fruit and vegetable intake most (11 out of 17) had a primary focus to change fruit and vegetable intake. There are several other reasons why positive changes were not identified for some dietary outcomes; young adults may identify health problems to be distal and therefore do not perceive a need to change their eating habits [89]. Also young adults have a number of competing time demands which may be prioritised over making changes to their eating habits (e.g., study, work, socialising, relationships, family obligations and /or parenthood) [90]. As a result, establishing which BCTs in those interventions are successful is especially crucial among this group. 

### 4.2. Effectiveness of BCTs

This review compared the individual BCTs in effective versus noneffective interventions to identify which specific BCTs may be agents of change. Thirteen BCTs had an effectiveness ratio >50% and ‘habit formation’ had the highest percentage effectiveness ratio (five out of five, or 100%). An example of this technique in an intervention includes; prompting participants to fill half of their plate with either salad or cooked vegetables each night at dinner. These results correspond with another meta-analysis which found habits to be consistently correlated with physical activity and nutrition behaviour [91]; with a medium-to-large grand weighted mean habit–behaviour correlation (r_+_ ≈ 0.45). Previous work has suggested that habits are behavioural patterns that are learned through context-dependent repetition [92,93], and future research should focus on promoting consistent repetition of behaviour in unvarying contexts to ultimately reinforce context–behaviour associations and automatically cue the habitual response [94].

Focusing on the natural consequences of performing a behaviour was found to be effective; with ‘information about health consequences’ demonstrating a percentage effectiveness ratio of 52% (effective in 15 out of 29 studies). An example of this technique in an intervention includes providing participants with morbidity and mortality statistics from related health conditions. Emphasising the consequence by making it more memorable had even greater effectiveness with ‘salience of consequences’ demonstrating an effectiveness ratio of 83% (effective in five out of six studies). An example of this techniques used in an intervention includes; presenting visual images of the negative consequences of eating unhealthily, i.e., blocked arteries. These results are comparable to a review of 45 brief nutrition interventions in adults, whereby accentuating the consequence of a behaviour had greater effectiveness than just providing information about a health consequence [29]. In addition, the World Health Organization has shown that the strong emphasis on the consequence has been particularly successful in changing tobacco smoking behaviours, with graphic warnings present on cigarette packets helping to reduce intention to smoke across many countries [95]. Also this approach was particularly effective among young adults [96]. As such, there is potential for a similar approach to change nutrition behaviours with graphic warnings present on energy-dense nutrient-poor foods such as sugar sweetened beverages. 

Adding objects to the environment was a technique that was shown to be effective with a percentage effectiveness ratio of 70% (effective in seven out of 10 studies). An example of this in an intervention was to provide participants with a free fruit and vegetable box to facilitate healthy eating. The effectiveness of this technique may coincide with known financial barriers associated with this age group [97]. These results correspond with another review in overweight and obese adults, which showed that adding objects to the environment significantly (*p* < 0.04) predicted long-term effect on diet and physical activity behaviours [25].

Furthermore, setting goals (effective in 18 out of 32 studies or 56%) and planning actions (effective in 16 out of 27 studies or 59%) in terms of performing a target behaviour were found to be effective BCTs. Examples of these techniques in interventions were (i) participants set a goal of eating two pieces of fruit per day and (ii) participants were instructed to write plans of when and where they would consume an extra piece of vegetable per day, for the next week. The effectiveness of goal setting of behaviour is comparable to a meta-regression analyses among 48 interventions in overweight and obese adults, which showed that this particular BCT was independently associated with better intervention effects on diet and physical activity behaviours at short (*p* < 0.001) and long term (*p* < 0.01) [25]. Similarly, the effectiveness of action planning resembles the findings in another review of brief nutrition interventions in adults which demonstrated a percentage effectiveness ratio of 59% [29].

The current review established no difference in number of BCT’s in relation to effectiveness on nutrition behaviours. This corresponds with several other systematic reviews in different population groups [29,98]. Overall, findings suggest that a greater number of BCTs is not necessarily more efficacious, and the utility of the individual BCTs may be more important.

### 4.3. Strengths and Limitations of Included Studies 

The risk of bias assessment identified several strengths. Generally, studies provided adequate detail to describe study attrition, any exclusions from the analysis and suitable generation for the allocation sequence. Despite this, many studies either had high risk of bias or lacked the necessary detail to judge the quality for selective outcome reporting, allocation concealment and blinding of participants, personnel and/or outcome assessors. Additionally, publication bias was evident in the meta-analysis of fruit and vegetable intake. There was an over-representation of studies in primarily female populations, in Caucasian samples and within college/University settings, which may limit the generalisability. Studies primarily focused on a limited selection of food groups with most focusing on fruit and vegetable intake, but very few considered overall diet quality. Few studies measured long-term dietary outcomes; as such, there is limited evidence to determine the long-term effectiveness of dietary interventions in young adults. 

Coding of BCTs was problematic due to inconsistencies and differences in reporting of intervention components. A stringent coding strategy was applied, where BCTs were not coded as present if there was insufficient description. However, some of those BCTs coded as absent techniques may have been apparent but not adequately described. This issue has been identified in previous attempts to categorise intervention content [30], and therefore greater attempts must be made by authors to sufficiently describe this information by publishing intervention protocol papers and/or including a checklist of the exact BCTs utilised as Appendix A of each publication. 

### 4.4. Strengths and Limitations of Review

The novelty of this review in being the first to explore which BCTs are associated with effective interventions in young adults is a strength, as is the use of a robust method to code the BCTs, using the most recent and comprehensive taxonomy of BCTs available [18]. Although more evidence is required, this is an important first step towards distinguishing between effective and noneffective components of interventions and allowing for replicability of effective techniques in developing future interventions. Other strengths include use of a comprehensive search strategy, two independent reviewers at each stage of the review, robust statistical analysis and the use of the Cochrane Collaboration’s Tool for assessing risk of bias [23]. This review is limited by only including studies published in English language, and therefore relevant studies may have not been identified. To obtain a large volume of literature, a broad age range (aged 17–35 years) was used to define young adults. However, young adults are not a homogenous group [99]. Traditionally, young adulthood was thought to be the time from adolescence through until 40 years of age when middle adulthood begun [100]. However, emerging adults who are in their late teens/early 20s are likely to be very different to those in their late 20s/ early 30s in lifestyle factors affecting health such as marital status, occupational status, housing environment, educational attainment and family circumstances [101]. Therefore, future research should consider presenting results by age subgroups to better reflect the diversity in dietary and health behaviours. In determining the effectiveness of BCTs, a percentage effectiveness ratio was utilised. This approach has been implemented by similar reviews [29,30,102]. However, this approach uses a binary categorisation as effective or not effective but does not consider the size of the effect. Also, to identify a greater amount of literature, and since dietary outcome is rarely treated in isolation, the scope was expanded to include lifestyle behaviours interventions (focusing on diet, PA and/or treating or preventing obesity). However, in those studies where dietary change was not a primary outcome they were likely to be underpowered to detect statistically significant changes in diet. Hence, only the most robust of changes in diet will have been detected in the current evidence synthesis.

## 5. Conclusions

This systematic review and meta-analysis highlights the potential of lifestyle behavioural interventions to improve young adults’ fruit and vegetable intake, but the evidence was less convincing for other dietary outcomes. More high quality studies are needed to determine longer-term effectiveness, and with consideration of overall diet quality. This is especially important due to the poor eating habits of young adults worldwide. This review also demonstrated the potential of the following BCTs; ‘habit formation’, ‘salience of consequences’ and ‘adding objects to the environment’ in positively changing dietary behaviour. However, due to the lack of studies, including each BCT, this review could not identify which BCTs are imperative to success and more studies are required before confirming which BCTs can be considered as having greater effectiveness than others. To better understand which BCTs are linked to intervention efficacy, future research must describe interventions in detail to allow for identification and replication of information, for example, publishing intervention protocol papers and/or including a checklist of exact BCTs implemented with supplementary material of each publication. Identifying and utilising effective BCTs for dietary interventions in young adults will aid the development of potentially more effective and replicable interventions, policies and nutritional guidelines, thus building a strong evidence base to support healthy eating habits among young adults.

## Figures and Tables

**Figure 1 nutrients-11-00825-f001:**
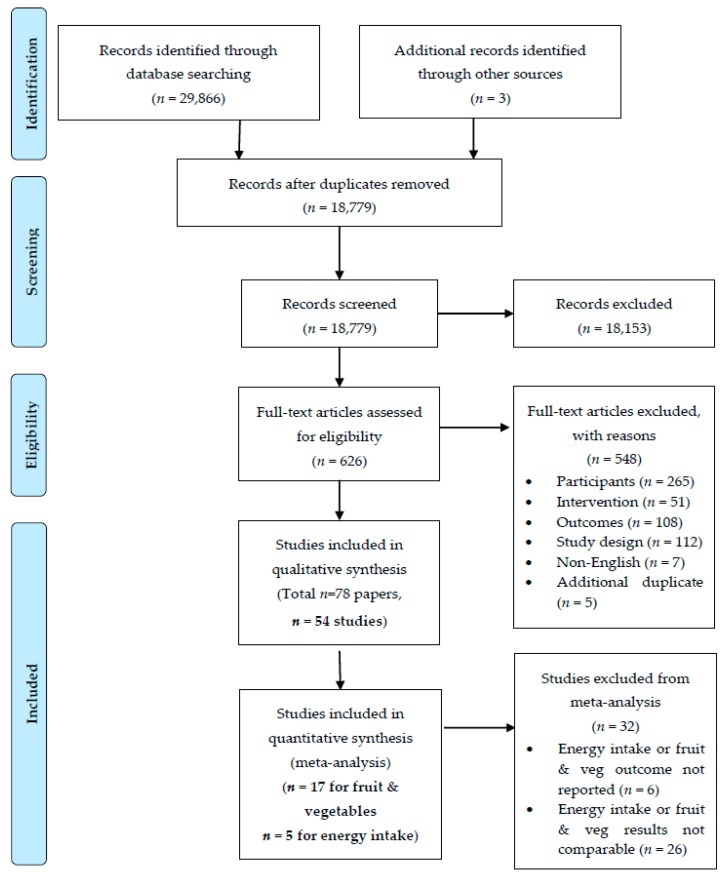
PRISMA flow diagram of included studies.

**Figure 2 nutrients-11-00825-f002:**
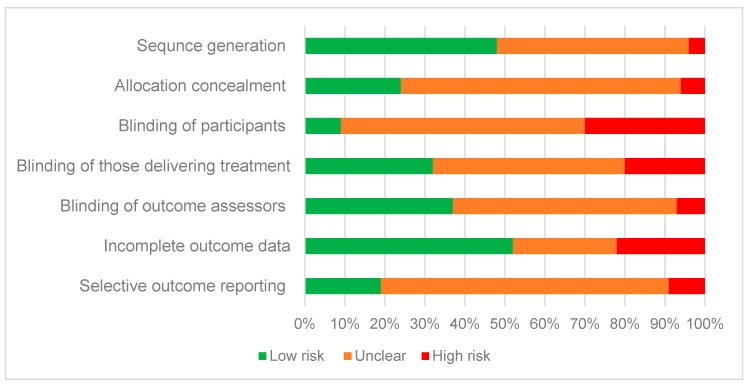
Percentage of studies from risk of bias assessment that were categorised as low, high or unclear risk for individual risk components.

**Figure 3 nutrients-11-00825-f003:**
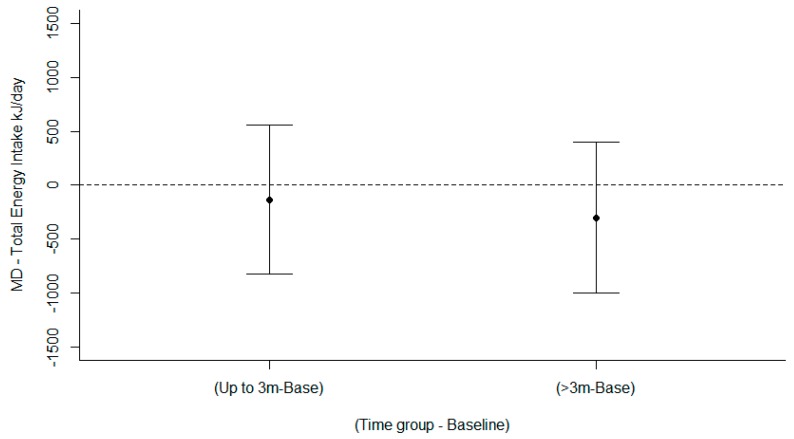
Mean differences between arms in Total Energy intake (kJ/day) over time.

**Figure 4 nutrients-11-00825-f004:**
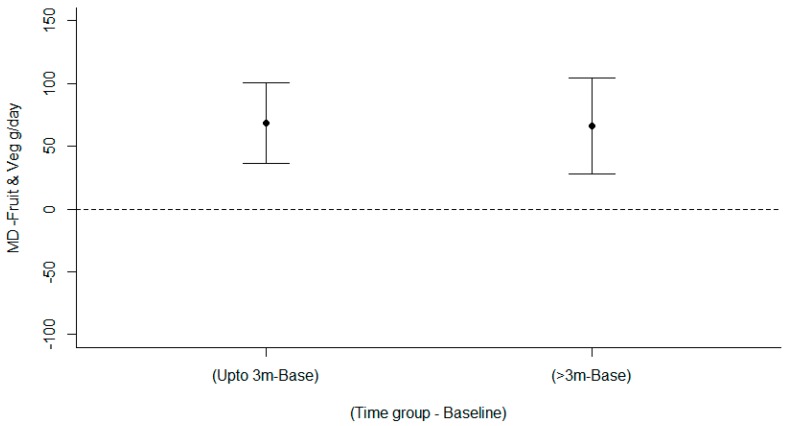
Mean differences between arms in fruit and vegetable intake (g/day) over time.

**Figure 5 nutrients-11-00825-f005:**
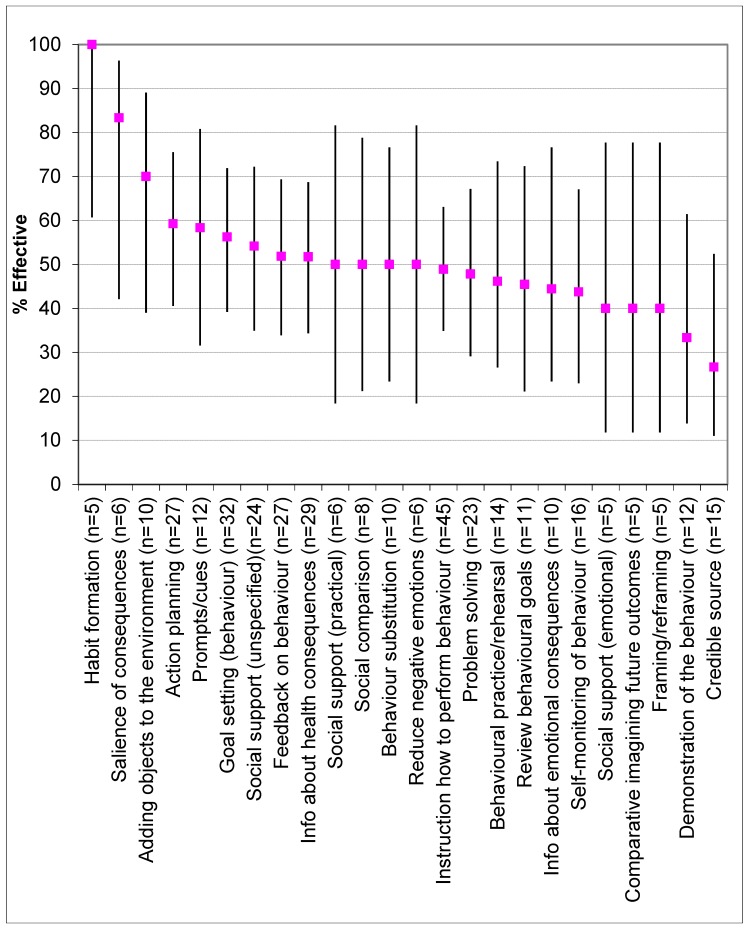
Percentage effectiveness of behaviour change techniques.

**Figure 6 nutrients-11-00825-f006:**
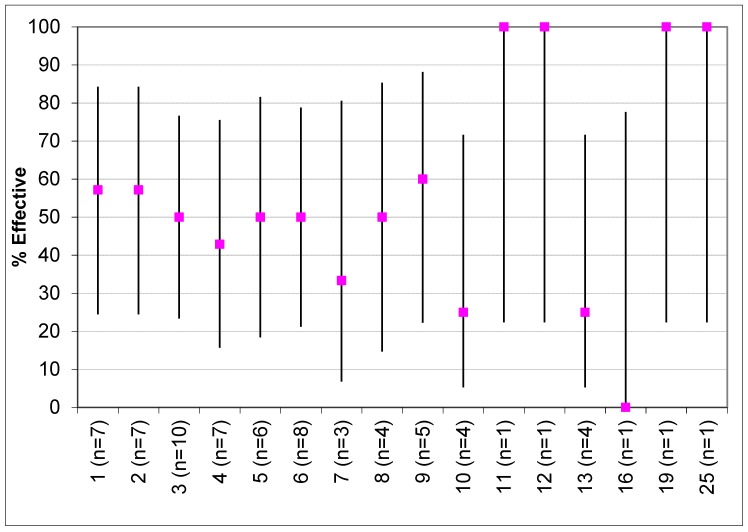
Percentage effectiveness of interventions by number of behaviour change techniques.

**Table 1 nutrients-11-00825-t001:** Summary of study characteristics from 54 interventions with dietary outcome in young adults.

		Total
Publication year	Before 2004 n (%)	2 (3.7%)
	2004–2008 n (%)	8 (14.8%)
	2009–2013 n (%)	15 (27.7%)
	2014–October 2018 n (%)	29 (53.7%)
Country	United States n (%)	30 (55.5%)
	Australia n (%)	9 (16.7%)
	Canada n (%)	3 (5.5%)
	UK n (%)	2 (3.7%)
	Other n (%)	10 (18.5%)
Number of participants	Total	16,383
	Mean	303.4
	Median	161.5
	Range	37 to 2343
Sex	Female only studies n (%)	11 (20.4%)
	Male only studies n (%)	4 (7.4%)
	Studies with both males and females n (%)	39 (72.2%)
	Average proportion of males in gender-neutral programs %	32%
Age	Mean years	20.9
	17–≤25 years n (%)	31 (57.4%)
	17–≤30 years n (%)	12 (22.2%)
	17–≤35 years n (%)	11 (20.4%)
Ethnicity	Predominantly white n (%)	32 (59.3%)
	Predominantly non-white n (%)	6 (11.1%)
	Not reported n (%)	16 (29.6%)
Education	Predominantly high school or less n (%)	1 (1.9%)
	Current University/ college students’ n (%)	32 (59.3%)
	Not reported or unclear n (%)	21 (38.9%)
Mode of intervention delivery	eHealth only n (%)	16 (29.6%)
	Face-to-face only n (%)	16 (29.6%)
	Print only n (%)	1 (1.9%)
	Multicomponent n (%)	21 (38.9%)
Primary outcome	Nutrition	32 (59.3%)
	PA	3 (5.6%)
	Weight or BMI (Obesity prevention focus)	11 (20.4%)
	Weight, BMI or waist circumference (obesity treatment focus)	8 (14.8%)
Dietary assessment	FFQ	25 (46.3%)
	Specific nutrient/food/diet behaviour questionnaire	17 (31.5%)
	Food record	5 (9.3%)
	Recall (24-hr, 3-day and 7-day)	5 (9.3%)
	Multiple	2 (3.7%)
Setting	College/University n (%)	37 (68.5%)
	Community n (%)	15 (27.8%)
	Military n (%)	2 (3.7%)
Study arms	Total	133
	2 arms n (%)	38 (70.4%)
	3 arms n (%)	10 (18.5%)
	4 arms n (%)	4 (7.4%)
	5 arms n (%)	1 (1.9%)
	6 arms n (%)	1 (1.9%)
Intervention duration	Mean duration (months)	4.2
	Range	Single session to 24 months
	Single session – ≤3-months n (%)	40 (74.1%)
	4–≤6-months n (%)	8 (14.8%)
	7–>12-months n (%)	6 (11.1%)
Length of follow-up from end of intervention	Mean length (months)	1.9
	Range	0 to 23-months
	No follow-up – ≤3-months n (%)	45 (83.3%)
	4–≤6-months n (%)	3 (5.6%)
	7–>12-months n (%)	6 (11.1%)
Retention rate	Post-intervention (mean %)	77.7%
	Range	22% to 98%
	At longest follow-up point (mean %)	66.2%
	Range	11% to 97%

**Table 2 nutrients-11-00825-t002:** Behaviour change techniques used ^a^.

Behaviour Change Technique	N	%
1.1 Goal setting (behaviour)	32	45.7%
1.2 Problem solving	23	32.9%
1.4 Action planning	27	38.6%
1.5 Review behavioural goals	11	15.7%
1.6 Discrepancy between behaviour and goals	1	1.4%
1.8 Behavioural contract	2	2.9%
1.9 Commitment	2	2.9%
2.1 Monitoring of behaviour by others without feedback	1	1.4%
2.2 Feedback on behaviour	27	38.6%
2.3 Self-monitoring of behaviour	16	22.9%
2.4 Self-monitoring of outcome(s) of behaviour	4	5.7%
2.6 Biofeedback	2	2.9%
2.7 Feedback on outcome(s) of behaviour	2	2.9%
3.1 Social support (unspecified)	24	34.3%
3.2 Social support (practical)	6	8.6%
3.3 Social support (emotional)	5	7.1%
4.1 Instruction on how to perform the behaviour	45	64.3%
4.2 Information about antecedents	3	4.3%
4.4 Behavioural experiments	1	1.4%
5.1 Information about health consequences	29	41.4%
5.2 Salience of consequences	6	8.6%
5.3 Information about social and environmental consequences	1	1.4%
5.6 Information about emotional consequences	10	14.3%
6.1 Demonstration of the behaviour	12	17.1%
6.2 Social comparison	8	11.4%
6.3 Information about others’ approval	2	2.9%
7.1 Prompts/cues	12	17.1%
8.1 Behavioural practice/rehearsal	14	20.0%
8.2 Behaviour substitution	10	14.3%
8.3 Habit formation	5	7.1%
8.4 Habit reversal	4	5.7%
8.7 Graded tasks	4	5.7%
9.1 Credible source	15	21.4%
9.2 Pros and cons	4	5.7%
9.3 Comparative imagining of future outcomes	5	7.1%
10.1 Material incentive (behaviour)	1	1.4%
10.2 Material reward (behaviour)	1	1.4%
10.3 Non-specific reward	1	1.4%
10.9 Self-reward	1	1.4%
10.10 Reward (outcome)	1	1.4%
11.2 Reduce negative emotions	6	8.6%
12.1 Restructuring the physical environment	4	5.7%
12.3 Avoidance/reducing exposure to cues for the behaviour	4	5.7%
12.5 Adding objects to the environment	10	14.3%
12.6 Body changes	1	1.4%
13.2 Framing/reframing	5	7.1%
13.3 Incompatible beliefs	1	1.4%
13.4 Valued self-identify	2	2.9%
13.5 Identity associated with changed behaviour	3	4.3%
15.1 Verbal persuasion about capability	2	2.9%
15.2 Mental rehearsal of successful performance	2	2.9%
15.3 Focus on past success	3	4.3%
15.4 Self-talk	2	2.9%
16.2. Imaginary reward	2	2.9%
16.3 Vicarious consequences	3	4.3%

^a^ Median number of BCTs used in interventions = 5; Range: 1–25.

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
