# Peer review of "Effectiveness of Interventions and Behaviour Change Techniques for Improving Dietary Intake in Young Adults: A Systematic Review and Meta-Analysis of RCTs"

_nutrients, 2019, doi:10.3390/nu11040825_

Reviewer 1 Report

This review contains novel information as it appears to be the first study to explore which BCTs are associated with effective interventions in young adults. This is a very large review; however, the authors have done a great job at compiling, analysing and interpreting the results. Overall, it is a very well written paper.

Author Response

We would like to thank the reviewer for their positive feedback for the manuscript. No changes have been made to the manuscript

Reviewer 2 Report

I have reviewed the paper entitled "Effectiveness of interventions and behavior change techniques for improving dietary intake in young adults: a systematic review and meta-analysis of RCTs." Overall, this paper is interesting, well organized and the authors explain the novelty of this analysis very well. The authors reviewed RCTs which examine behavioral interventions aimed at changing young adults eating behaviors and provide specific analysis on which types of behavioral interventions were most effective.

One concern I have is the reliance on the "effectiveness ratio" which examines the number of studies deemed to have an effect as compared to the total number of studies. It may help the reader to understand the size of these effects rather than simply using a binary categorization. Commentary on the size of the effect would also be beneficial to the reader to help contextualize the findings.

Secondly, the authors explain that a wide range of age was used to match the international definition of adulthood, and that included studies spanned multiple countries. Is there any consideration given to separate analyses of older vs. younger adults? Young adults who are currently in high school and living with their parents may have different responses and different access to interventions as compared to those studying at university or those who are living on their own.

Author Response

We would like to thank the reviewers for their critique of the manuscript. We believe the suggested changes have greatly improved the quality of the manuscript.

 Each comment has been considered by the authors, and we have addressed in turn. The manuscript has been revised to incorporate appropriate changes.

Point 1: One concern I have is the reliance on the "effectiveness ratio" which examines the number of studies deemed to have an effect as compared to the total number of studies. It may help the reader to understand the size of these effects rather than simply using a binary categorization. Commentary on the size of the effect would also be beneficial to the reader to help contextualize the findings.

 Response 1: The ‘percentage effectiveness ratio’ was implemented to coincide with other similar reviews (see refs below).

 ·         Martin, J.; Chater, A.; Lorencatto, F. Effective behaviour change techniques in the prevention and management of childhood obesity. International journal of obesity 2013, 37, 1287.

·         Michie, S.; Jochelson, K.; Markham, W.A.; Bridle, C. Low income groups and behaviour change interventions: a review of intervention content, effectiveness and theoretical frameworks. Journal of Epidemiology and Community Health 2009, jech. 2008.078725.

·         Whatnall, M.C.; Patterson, A.J.; Ashton, L.M.; Hutchesson, M.J. Effectiveness of brief nutrition interventions on dietary behaviours in adults: A systematic review. Appetite 2018, 120, 335-347.

 Adding in the size of the effect for each of the BCT’s would substantially increase the length of the paper. For example if a commentary was provided on only the top 5 BCT’s this would require a commentary on the effect for 40 studies. We have acknowledged this as a limitation of the study (Page 22, lines 709-712)

 “In determining the effectiveness of BCTs, a percentage effectiveness ratio was utilised. This approach has been implemented by similar reviews [28, 29, 98]. However, this approach uses a binary categorisation as effective or not effective but does not consider the size of the effect.

Point 2: Secondly, the authors explain that a wide range of age was used to match the international definition of adulthood, and that included studies spanned multiple countries. Is there any consideration given to separate analyses of older vs. younger adults? Young adults who are currently in high school and living with their parents may have different responses and different access to interventions as compared to those studying at university or those who are living on their own.

Response 2:

Thank you for this suggestion. While we agree there is heterogeneity in demographic characteristics which may influence diet between those classed as emerging adults (17-25yrs) vs those older (26-35yrs) we do not think this is possible with the included studies in this review. The majority of studies (n=31, 57%) included samples aged 17-25 years. Of the remaining 23 studies which considered a broader age range none of these specifically stratified results by emerging adults vs those older. There is a need for future research to acknowledge the differences between emerging adults and those young adults who are 25 years and beyond. This has been highlighted in the discussion section (Page 22, Line 702-709)  

 “To obtain a large volume of literature, a broad age range (aged 17-35 years) was used to define young adults. However, young adults are not a homogenous group [98]. Traditionally, young adulthood was thought to be the time from adolescence through until 40 years of age when middle adulthood begun [99]. However, emerging adults who are in their late teens/early 20s’ are likely to be very different to those in their late 20’s/ early 30’s in lifestyle factors affecting health such as; marital status, occupational status, housing environment, educational attainment and family circumstances [100]. Therefore, future research should consider presenting results by age sub-groups to better reflect the diversity in dietary and health behaviours.”

Please see attached file for a point-by-point response
